# Pathogenesis and Preventive Tactics of Immune-Mediated Non-Pulmonary COVID-19 in Children and Beyond

**DOI:** 10.3390/ijms232214157

**Published:** 2022-11-16

**Authors:** Hsin Chi, Lung Chang, Yen-Chun Chao, Dar-Shong Lin, Horng-Woei Yang, Li-Chih Fang, Chia-Hsueh Lin, Che-Sheng Ho, Kuender D. Yang

**Affiliations:** 1MacKay Children’s Hospital, Taipei 103, Taiwan; 2Department of Medicine, MacKay Medical College, Sanzhi, New Taipei City 252, Taiwan; 3Departments of Pediatrics and Medical Research, MacKay Memorial Hospital, TamSui, New Taipei City 251, Taiwan; 4Institute of Clinical Medicine, National Yang Ming Chiao Tung University, Taipei 112, Taiwan; 5Department of Microbiology & Immunology, National Defense Medical Center, Taipei 114, Taiwan

**Keywords:** coronavirus disease 2019 (COVID-19), meningoencephalitis (ME), acute necrotizing encephalopathy (ANE), multisystem inflammatory syndrome in children (MIS-C), non-pulmonary COVID-19, immunopathogenesis

## Abstract

The COVID-19 pandemic has evolved to immune escape and threatened small children and the elderly with a higher severity and fatality of non-pulmonary diseases. These life-threatening non-pulmonary COVID-19 diseases such as acute necrotizing encephalopathies (ANE) and multisystem inflammatory syndrome in children (MIS-C) are more prevalent in children. However, the mortality of multisystem inflammatory syndrome in adults (MIS-A) is much higher than that of MIS-C although the incidence of MIS-A is lower. Clarification of immunopathogenesis and genetic susceptibility of inflammatory non-pulmonary COVID-19 diseases would provide an appropriate guide for the crisis management and prevention of morbidity and fatality in the ongoing pandemic. This review article described three inflammatory non-pulmonary COVID-19 diseases including (1) meningoencephalitis (ME), (2) acute necrotizing encephalopathies (ANE), and (3) post-infectious multisystem inflammatory syndrome in children (MIS-C) and in adults (MIS-A). To prevent these life-threatening non-pulmonary COVID-19 diseases, hosts carrying susceptible genetic variants should receive prophylactic vaccines, avoid febrile respiratory tract infection, and institute immunomodulators and mitochondrial cocktails as early as possible.

## 1. Introduction

Coronavirus disease 2019 (COVID-19) has evolved from initial waves of mild pulmonary diseases to the later waves of potentially fatal diseases of non-pulmonary encephalopathies, and multisystem inflammatory syndrome in children (MIS-C) [1,2,3,4]. The evolutional variants of SARS-CoV-2 have caused immune escapes of vaccines and monoclonal antibodies, resulting in incomplete herd immunity and periodic re-emergence [5,6]. The dynamic interactions among the herd immunity, virus variants, and environments such as vaccination coverage, availability of monoclonal antibodies and antiviral medications, and quarantine strategies have shaped the features of the COVID-19 pandemic in different waves, showing a change from the initial higher fatality and lower transmission to a lower fatality and more contagious [7,8]. However, the lower vaccination coverage and limitation of anti-virus medication together with evolution of viral variants and lack of pre-existing immunity of small children have made children susceptible to infection and vulnerable to higher hospitalization rate [1,2,3,4,9,10,11].

The first notice for the threat of Omicron variants on children was raised by Ledford in December 2021 after a surge of pediatric hospitalization in South Africa [12]. The increase of pediatric hospitalization was also later found in Europe, and higher morbidity and fatality of pediatric encephalopathies are found in Hong Kong and Taiwan [13,14,15]. In Asia, there was no COVID-19-related death of children reported before October 2021 in Korea, January 2022 in Hong Kong, February 2022 in Singapore, or April 2022 in Taiwan [14,15,16,17]. The Omicron BA.2 pandemic appeared to be more neuropathogenic in children than the earlier waves of Alpha and Delta COVID-19 pandemic [14,15]. 

The non-pulmonary encephalopathies and MIS-C in the later wave of Omicron subvariants have caused a high morbidity and mortality in children [14,15,16,17]. For instance, there have been more than 30 deaths of children reported in Taiwan during the Omicron COVID-19 outbreak between April and September 2022, of which more than one half died of encephalopathies [15]. More importantly, adolescents and adults could have non-pulmonary COVID-19 diseases such as acute necrotizing encephalopathies (ANE) [18,19,20], and multisystem inflammatory syndrome in adults (MIS-A) [21,22,23]. Although the incidence of ANE and MIS-A is lower in adolescents and adults, the mortality in MIS-A is much higher than that in MIS-C [21,22]. Based on immunopathogenesis of the pulmonary COVID-19 different from non-pulmonary COVID-19, this review article depicts the dynamic clinical features and immunopathogenesis on non-pulmonary COVID-19 diseases and provides immunopreventive tactics for early prevention and treatment of the potentially fatal hyperinflammatory non-pulmonary COVID-19.

## 2. Methodologies for Structured Literature Search on Non-Pulmonary COVID-19

In an observation of the dynamic change of COVID-19 in children evolving from early mild pulmonary disease to later fatal non-pulmonary COVID-19 in Asia [14,15,16,17], we structured the review article of non-pulmonary COVID-19 into 5 sections including (1) dynamic waves of non-pulmonary COVID-19, (2) immunopathogenesis, (3) predisposing factors, (4) immunopreventive tactics, and (5) conclusions. 

### 2.1. The Keywords for Collecting References for Depicting the Dynamic Waves of COVID-19

The keywords including COVID-19 and SARS-CoV-2 together with dynamic waves or virus variants were input to PubMed Center and publisher websites for biomedical literature. These references were layout to the first section “dynamic changes from pulmonary to inflammatory non-pulmonary COVID-19”. 

### 2.2. The Keywords for Collecting References for the Immunopathogenesis of Non-Pulmonary COVID-19

To define the immunopathogenesis of non-pulmonary COVID-19, we input COVID-19 or SARS-CoV-2 as an initial keyword which was combined with meningoencephalitis, encephalopathy, or multisystem inflammatory syndrome, and with mechanism or pathogenesis to collect references for the second section “immunopathogenesis of inflammatory non-pulmonary COVID-19 in children and adults”. To conquer the heterogeneity of diagnostic criteria of ME, ANE, and MIS-C, we included the references of ME defined with pleocytosis in CSF without focal cerebral lesion on the imaging study, included the references of ANE defined with the deep brain cell necrosis on the imaging study, and included the references of MIS-C defining the criteria with fever at least 4 days and inflammatory syndrome at least 2 systems after COVID-19 for two to six weeks.

### 2.3. The Keywords for Collecting References for Predisposing Factors of Non-Pulmonary COVID-19

To identify the predisposing factors of non-pulmonary COVID-19, we input COVID-19 or SARS-CoV-2 as an initial keyword, which was combined with meningoencephalitis, encephalopathy, or multisystem inflammatory syndrome; and with immunodeficiency, genetic association, metabolic disturbance, or mitochondrial dysfunction. 

### 2.4. Collecting References for the Immunopreventive Tactics of Non-Pulmonary COVID-19

To formulate the immunopreventive tactics for non-pulmonary COVID-19, we input COVID-19 or SARS-CoV-2 as an initial keyword, which was combined with active immunization, passive immunization, anti-virus replication, anti-inflammation, cytokine storm, or mitochondria cocktails. 

Taking together these structured references, we sketched a figure to address the dynamic changes from pulmonary to non-pulmonary COVID-19 in terms of transmission and clinical features of pneumonia, encephalopathy, or MIS-C (Figure 1). We summarized the pathogenesis of non-pulmonary COVID-19 diseases into a structured table describing different immunopathogenesis, infection-associated hyperinflammation, race and inheritance, and immunotherapies for non-pulmonary COVID-19 (Table 1). Finally, we drew a figure depicting the immunopreventive tactics for non-pulmonary COVID-19 by active immunization, passive immunization, anti-leukocyte activation, anti-cytokine storm, and correction of metabolism, which are different from the conventional anti-virus medication for patients with pulmonary COVID-19. 

## 3. Dynamic Changes from Pulmonary to Inflammatory Non-Pulmonary COVID-19

An emerging novel viral infection usually arises from microbial mutation or zoonosis that causes cross-species human to human transmission [24]. The emerging infection progresses from endemic to epidemic to pandemic depending on dynamic interactions among herd immunity, viral virulence, personal protection equipment and environmental lockdowns. Taking an influenza pandemic as an example, a novel cross-species influenza virus that causes human to human transmission could raise a pandemic over 30% of the population in the initial years because almost all humans are susceptible to the novel influenza virus [25]. When an emerging variant of the influenza is introduced into a susceptible population, it usually causes a pandemic with an attack rate of 10–30%, and several re-emerging waves will follow until herd immunity over 60% is reached [26].

The COVID-19 pandemic has caused different waves of outbreaks from Alpha, Beta, Delta, and Omicron variants [7,8,9,10,11,12,13,14,15,16,17,24,27,28,29], similar to a novel influenza pandemic [25,26], which does not cease the spreading until 60–70% of population are affected by viral mutants.

In the beginning of COVID-19 pandemic, the transmission rate in children was much lower than that in adults, but the transmission rates have surged dramatically in the latest wave of Omicron pandemic (Figure 1A). Comparing to previous ancestors, Alpha, Beta, and Delta variants, the new Omicron variants demonstrated mild upper respiratory illness (Figure 1B), but severe non-pulmonary diseases, including multisystem inflammatory syndrome in children (MIS-C) (Figure 1C) and neurological involvement emerged prominently in children (Figure 1D). It is believed that the dominant Omicron variants will eventually affect most of the population, particularly infants and children are more vulnerable to severe diseases due to risk factors of immature immunity, low rate of vaccination coverage, and the social behavior on everything cosseting and something vulnerable to breakthrough [27,28,29,30,31,32].

Nevertheless, the Omicron variants have evolved into BA.1, BA.2, BA.3, BA.4, and BA.5 subvariants associated with immune escape [31,32]. Interestingly, the severity of MIS-C has significantly decreased over the pandemic waves of Alpha, Beta, Delta, and Omicron [33,34,35,36]. This trend may be not applicable to other countries due to diverse virus variants, and availability of vaccines and anti-virus drugs. In Asian countries where strict quarantine regulations such as enforcement of face mask and maintaining social distancing, the pandemic in children was skewed toward later transmission with Omicron variants [14,15,16,17]. In fact, children are susceptible to infections with Omicron variants at home where exposures were higher than transient exposures at school [12].

A higher hospitalization rate for children with infection of Omicron variants was initially reported from Gauteng Province, South Africa, where 462 (18%) of hospitalizations were patients less than 19 years of age, higher than those in the three previously pandemic waves [32]. In two larger cohort studies of COVID-19 in children found that hospitalized children aged < 2 years or with comorbidities were susceptible to severe COVID-19 and potential fatality [1,2,33,34]. Higher severity and fatality of non-pulmonary disease have been later found in Hong Kong and Taiwan where the Omicron BA.2 was prevalent [14,15]. Fortunately, COVID-19 mRNA vaccines have been authorized for children over 6 months of age. However, it is also concerning that children and adolescents getting an infection within a month after first dose of COVID-19 vaccination might be susceptible to myocarditis, which is comparable to a higher rate of myocarditis occurring to children and adolescents with the second dose of COVID-19 vaccination [27]. Although the severity of pulmonary COVID-19 had significantly decreased in the Omicron pandemic [1,2], however, the non-pulmonary COVID-19 diseases such as ANE [18,19] and MIS-A [21,22] revealed a high morbidity and mortality.

## 4. Immunopathogenesis of Inflammatory Non-Pulmonary COVID-19 in Children and Adults

Along the different waves of COVID-19 pandemic from Alpha, Beta, Gamma, to Delta variants, the MIS-C prevalence significantly decreased [35,36]. However, in the current wave of Omicron variants, higher transmission and hospitalization of children were prominently complicated with non-pulmonary diseases [37,38,39,40], especially meningoencephalitis, ANE, and post-infectious autoimmune diseases such as MIS-C and acute disseminated encephalomyelitis (ADEM) were noticed in the late waves of pandemic [39,40,41,42,43,44,45]. Moreover, adolescents and adults are not spared from ANE or MIS in adults (MIS-A). Therefore, everyone needs to know that non-pulmonary COVID-19 could occur to children, adolescents, and adults. Understanding the clinical features and immunopathogenesis of non-pulmonary COVID-19 in children and adults for early diagnosis and early treatment is crucial to prevent morbidity and mortality. We here presented three non-pulmonary COVID-19 diseases and tried to delineate the underlying immunopathogenesis for early recognition and prevention of potential fatal non-pulmonary COVID-19 diseases.

### 4.1. Immunopathogenesis of Meningoencephalitis

In the initial outbreak of COVID-19, the attack rate of children below 19 years of age were relatively lower, but higher rate of post-infectious autoimmune disease, MIS-C [1,2,3,4,36,46]. Sporadic cases of the COVID-19 children with neurological manifestations or encephalitis were reported with favorable recovery, no matter if those with detectable virus had RNA in CSF or not [47,48]. Common neurological manifestations of COVID-19 in children are conscious change, behavior change, vomiting, and/or seizure [39,40]. The risk factors in infants and children, contributing to severe COVID-19, included premature infants and children less than 2 years of age, who have immature immunity, and low levels of immunoglobulin A and G production. In addition, children with comorbidities such as immunodeficiency, cerebral palsy or obesity are also risk factors to severe COVID-19 [33,34,46,47,48,49].

COVID-19 associated with ME has been correlated to different mechanisms including virus invasion into the brain via olfactory nerve or dissemination through hematogenous, and immune-mediated cytokine storm or COVID-19-associated leukocyte activation, which may interrupt the blood–brain barrier or induce inflammation. Clarification of the virus or immune mediated mechanism is important for the therapeutic strategies with anti-virus or anti-inflammation regimen. It remains unclear whether COVID-19 children with ME are related to viral invasion or immune inflammation although immune-mediated inflammation is preferred. In the literature review, there were limited cases of COVID-19 children with ME or neurological manifestations found to have virus invasion in CSF or brain [39,40,47,48,49,50], suggesting immune-mediated mechanism is the major cause.

### 4.2. Immunopathogenesis of ANE

Several pediatric encephalopathies of COVID-19 have been reported in the literature, including meningoencephalitis, status epilepticus, Guillain–Barré syndrome (GBS), ADEM, but not ANE [39,40,47,48,49,50]. ANE is not a new disease but mostly occurs in children with an infectious disease, such as influenza, rotavirus, enterovirus, or herpes virus [51,52,53,54,55,56]. This disease is more frequently reported from Asian countries, especially related to influenza associated ANE, which is usually associated with a sudden onset of high fever, intractable seizure, and progress to coma with a high fatality in a few days [51,52]. In Caucasians, ANE is related to RAN binding protein 2 (RANBP2) gene mutations, called ANE type 1 (ANE1), which could be familial and recurrent [53,54]. Interestingly, ANE is rarely identified in the early waves of COVID-19 pandemic [39,40]. In a large cohort study, the neurological symptoms in pediatric COVID-19 were mostly transient (88%), and the life-threatening encephalopathies were 12%, in which most of the patients were MIS-C patients [40]. The fatalities of neurological diseases of COVID-19 in children were 2 to 3% [39,40]. Although the ANE was not described in the large cohorts of COVID-19, but sporadic case reports associated with COVID-19 in children were reported with favorable outcomes [39,40,55,56]. In contrast, a severe form of acute hemorrhagic necrotizing encephalopathy (AHNE) associated with COVID-19 in adults is more frequently reported with fatal outcomes [18].

The RANBP2 gene mutations have been associated with the ANE in children [53,54], some other genes regarding metabolism or immunity, including neuronal sodium channel alpha 1 (SCNIA) and carnitine palmitoyltransferase II (CPT2), human thiamine transporter 2 (hTHTR2) have been also linked with the ANE after an infection [55,57,58]. In addition, HLA genotypes (DRB1*09:01, DQB1*03:03) were also correlated with ANE in Japan [59]. The ANE associated with mutations of CPT2 and hTHTR2 are often sensitive to febrile illness including febrile infections [55,57]. The ANE is mostly found in children, but adults are not spared [19,20]. ANE in children is prevalent in Asian countries, including China [60], Japan [61,62], and Taiwan [62,63], and most of the ANE in children are induced by an infection, particularly influenza [60,61,62,63]. Early intervention of plasma exchange [60], immunomodulation [62,63], or hypothermia [64] for the ANE was recommended to limit mortality and morbidity. Since malignant fever and cytokine storm were prominent in ANE, hypothermia [64] or plasma replacement [60] in addition to IVIG and methylprednisolone pulse therapy have also been proposed to rescue the poor outcomes [60,61,62,63]. 

Further studies to identify genetic and mechanistic biomarkers are needed to prevent the patients with ANE from mortality and morbidity. For those with genetic variants of CPT2 and hTHTR2 leading to metabolic encephalopathies, early supplementation of mitochondrial cocktails such as biotin, thiamine, CoQ10, and L-carnitine might rescue the morbidity and mortality of ANE [53,54,55]. For those who possess genetic variants of RANBP2, SCNIA, or HLADRB1*0901 contributing to infection-associated hyperinflammation, early administration of immunomodulation and/or hypothermia therapy is required [54,58,59].

### 4.3. Immunopathogenesis of MIS-C

There are many post-COVID-19 infectious immunological diseases have been reported in the literature, such as MIS-C, ADEM, GBS, and transverse myelitis [39,40]. MIS-C is a typical post-COVID-19 autoimmune disease because it occurs between 2 and 6 weeks after COVID-19 infections. The clinical features of MIS-C are like those of KD, showing prolonged fever, skin rashes, fissure lips, non-purulent conjunctivitis, and cardiovascular involvement [65,66]. We have summarized the different phenotypes, laboratory data and immune mechanisms between KD and MIS-C, and shown that MIS-C patients had the older age, a history of COVID-19 two to six weeks before, coagulopathy with higher D-dimer levels and thrombocytopenia [66]. We have taken an opportunity to compare the different cytokine and chemokine profiles in blood between KD and MIS-C in Figure 2, showing that the IL-12 and IFNγ levels in MIS-C were much higher than those in KD; vice versa, the IFNα and IP-10 levels were much lower in MIS-C than in KD. This suggests that there should be certain different immune alterations contributing similar phenotypes between both diseases.

In fact, clinical features of MIS-C are closer to those of KD shock syndrome (KDSS), which are prominently found in female Hispanos with older age, and more frequently associated with thromboembolism and shock syndrome [65,66,67]. Fortunately, the fatality of MIS-C has decreased from 2% down to 0.6% in term of early recognition and treatment by IVIG and pulse methylprednisolone, and additional use of anakinra, anti-TNF or anti-IL-6 while refractory, or use of anticoagulants while manifestation of thromboembolic symptoms and signs [35,36,68,69]. The MIS variant in adults called MIS-A, although rare and atypical symptoms of MIS, revealed a higher fatality between 5 and 7% [21,22,23]. The MIS-A is also like another KD variant called atypical KD, which occurs to younger infants and older children, or even adolescents and adults, with higher cardiac complications and fatality [65,66,67]. This alerts clinicians that early diagnosis of MIS-C and MIS-A for early treatment should be kept in mind but not delayed diagnosis waiting for more criteria of definite diagnosis.

In the early waves of COVID-19 pandemic, MIS-C is prominently found in Western countries but not Asian countries [1,2,35,36]; however, there are more MIS-C patients found in Asian countries in the later waves of COVID-19 pandemic with Omicron variants [14,15,16,17]. Apparently, the prevalence of MIS-C was not only associated with races but also virus variants and different waves of pandemic under dynamic interactions between children and environments.

## 5. Predisposing Factors of Non-Pulmonary COVID-19 Diseases

Respiratory infectious diseases such as SARS-CoV-2, influenza and enterovirus infections could develop acute and post-infectious non-pulmonary diseases including ME, ANE, GBS, and ADEM. Recently, the incidence of MIS-C decreased in Western countries but increased in Asian countries [14,15,16,17,35,36,68,69]. Both MIS-C and MIS-A could also cause fatal outcomes due to post-infectious autoimmunity. Moreover, severe acute encephalopathies in adults have been sporadically reported in adults [18,19,20], and COVID-19 encephalopathies with fatal outcomes were prevalent in Asian countries [14,15]. Taken together, these non-pulmonary COVID-19 diseases tend to have predisposing factors related to (1) immature immunity, (2) metabolic disturbance linked to hyperinflammation, or (3) genetic association. 

### 5.1. Immature (Developmental) Immunity 

The younger the age the higher proportion of non-pulmonary COVID-19 severity has been described. Vaccine recipients are likely susceptible to breakthrough infections or reinfections due to a rapid decline of neutralizing antibody titers directed against different SARS-CoV-2 variants [70,71,72]. Moreover, only one third of BNT162b2 vaccine recipients and one quarter of COVID-19 patients had neutralizing antibody titers above the protecting titer of Omicron variants [70]. The neutralizing antibody titers against the Omicron variant after vaccination was substantially lower than those against the ancestral virus or the Beta variant.

We have previously shown small children had the higher viral load and longer shedding time of H1N1 influenza infection [73]. Children less than 5 years old are susceptible to enterovirus 71 (EV71) associated with delayed CD40L expression, which is involved in the switch of IgM to IgG production [74]. Viral meningoencephalitis such as EV71 usually occurs to children less than 5 years old [75]. In the COVID-19 pandemic, several studies have also demonstrated that the younger the age revealed higher viral load and longer virus shedding [76,77,78]. Recently, Torjesen reported that a steep rise in hospital admissions of very young children in COVID-19 Omicron pandemic [79]. More importantly, children with severe COVID-19 disease were found to have lower regional and systemic immune responses to SARS-CoV-2 [80]. Children with previous exposures to vaccines or earlier waves of COVID-19 tended to have lower serum neutralizing antibody titers directed against Omicron variants [70,72,81]. The immature immunity with low antibody titers, higher viral load and altered hyperinflammation in children made them vulnerable to severe non-pulmonary COVID-19 (Table 1).

### 5.2. Metabolic Disturbance Linked to Hyperinflammation 

There are some genetic backgrounds vulnerable to infection or fever by which innate immunity is linked to mitochondria dysfunction followed by proinflammatory response [82]. The fact that limited patients with COVID-19 encephalopathies had detectable SARS-CoV-2 in CSF or brain [39,40,47,48,49,50] suggested immune-mediated pathogenesis of COVID-19 encephalopathy is likely the major mechanism. This is supported by most of the studies demonstrating early immunomodulation of patients with encephalopathies could rescue the patients from mortality and morbidity [58,60,61,62,83]. This altered hyperinflammation in COVID-19 encephalopathies could be categorized into two potential mechanisms: (a) infection immunity induces inflammation that interrupted BBB for cerebral edema and inflammation, and (b) viral invasion to CNS results in inflammation. It is postulated that a neurotrophic viral infection could invade CNS via olfactory nerve or hematogenous dissemination that causes inflammation of blood brain barrier, neural cell apoptosis or necrosis [82]. This has been shown in enterovirus 71 neurotropism for motor neurons in the spinal cord and brainstem, responsible for encephalitis, showing pleocytosis of CSF, brain stem involvement, pulmonary hemorrhage, and edema [75]. However, there are very few studies demonstrating SARS-CoV-2 could cause neuronal cytotoxicity, and few studies showing direct evidence on the SARS-CoV-2 invasion of CNS or reporting anti-virus regimen rescued the children with encephalopathies. In contrast, certain ANE could be rescued by fever control, early immunomodulation, or early administration of vitamin cocktails for mitochondrial dysfunction [55,83,84,85,86].

Given the fact that patients with infection-associated ANEs usually have malignant fever and intractable seizure with hyperinflammation, we could postulate that an infection-induced fever and/or metabolic disturbance may be linked together to induce rapid hyperinflammation (Table 1). This is supported by two lines of evidence: (a) hypothermia therapy could effectively decreased temperature, lactic acidosis and inflammatory mediators in ANE patients, and (b) some reports showed early administration of carnitine, biotin, and thiamine could rescue patients with infection-induced ANE for favorable outcomes [55,84,85,86]. Taken together, it is believed that early recognition of genetic variants which are sensitive to fever and/or metabolic mitochondrial dysfunction with hyperinflammation may be subject to rescue of infection associated ANE by avoidance of fever, and correction of mitochondrial dysfunction (Table 1). 

### 5.3. Genetic Association of Severe Non-Pulmonary COVID-19 Diseases

Many genetic variants of immune genes and virus receptors have been associated with morbidity and mortality of certain emerging infections, including HIV, dengue, and coronaviruses [87,88,89,90,91]. It remains unknown what genetic variant(s) is(are) involved in the pathogenesis of COVID-19-associated ANE. The RANBP2 gene mutations increase the susceptibility to recurrent episodes of ANE with respiratory viral infections, particularly influenza infection [58,83]. RANBP2-associated ANE is more prominently found in Caucasians and called ANE1 [53,54]. Influenza-associated ANE is more prevalent in Asian countries than that in Western countries, but fewer cases were associated with RanBP2 mutation [53,54,60,61,62,83,85,92,93]. It is surprising that ANE children were not reported in some large cohorts of COVID-19 neurological involvement in the early waves of COVID-19 pandemic in Western countries, but were more prevalent in late waves of COVID-19 Omicron variants in Asian countries [14,15,16,17,39,40]. In addition to RanBP2 mutations, HLA, CPT2, SCN1A, and SLC19A3 mutations have been associated with influenza or COVID-19-associated ANE (Table 1) [53,55,57,59,93,94,95]. Interestingly, these genes associated with ANE are sensitive to infection, fever, and metabolic mitochondrial dysregulation of inflammation [56,57,58,94]. CPT2 is sensitive to fever associated with lower enzyme activity contributing to mitochondrial dysfunction with oxidative stress and inflammation [95,96]. Mutation of SCN1A is associated febrile seizure, and SLC19A3 is associated with defective thiamine transportation for mitochondrial energy supply via acetyl-CoA metabolism [55]. As mentioned above, hypothermia therapy or correction of biotin, thiamine, and carnitine could raise better outcomes of ANE [55,64,84,85,86]. Children carrying fever- or metabolic-sensitive genetic variants should receive prophylactic vaccines, encourage them to avoid respiratory tract infections, at least influenza and coronaviruses, and to actively control initial fever and administer biotin, CoQ10, vitamin B6, carnitine, and/or thiamine in 24 h [84,85,86,87].

The MIS-C is a new disease occurring between 2 and 6 weeks after a SARS-CoV-2 infection, suggesting a post-infectious autoimmune disease. The early cytokine storm profiles of MIS-C were similar, but certain different, to those found in KD (Figure 2) [65,66,97]. A study with 39 MIS-C patients showed that one quarter of MIS-C patients harbored heterozygous missense mutations in primary hemophagocytic lymphohistiocytosis (pHLH) genes (LYST, STXBP2, PRF1, UNC13D, AP3B1) or the HLH-associated gene DOCK8 (four variants) [98]. Patients with defects in SOCS1, XIAP, or CYBB exhibiting downstream activations of IL-18, oncostatin M, and nuclear factor κB were also reported in MIS-C patients [99]. In contrast, adult COVID-19 patients with MIS-A have been associated with autophagy genes (LGALS8, TECPR1), viral restriction factor genes (PLIN3, EXOSC5, RNASE2), and immune responses (ERAP1, SIGLEC15, GAB2, GOLGA4, SNX3) in addition to Kawasaki disease genes (PEAR1, ERAP1) [100].

In a multi-omics study with 76 MIS-C patients compared to 100 COVID-19 children, Sacco, et al. reported that the T cell clone carrying T cell receptor beta chain variant 11-2 (TRBV11-2) expression was prominently present in MIS-C patients rather than in non-MIS-C COVID-19 patients, and presence of auto-antibodies directed against several self-antigens have been reported in MIS-C patients [101]. This suggests the Vβ chain encoded by TRBV11-2 (Vβ21.3) strongly interacts with the superantigen-like motif of SARS-CoV-2 spike glycoprotein, mediating expansion of TRBV11-2 T cells. It is, however, debatable to recognize whether the S epitope of SASR-CoV2 associated with autoreactive T cell clone is mechanistic pathogenesis of the MIS-C. If the S epitope could induce autoreactive T cell expansion for MIS-C, the COVID-19 vaccines, such as Novavax, made in full length of Spike antigen would be also possible to induce the MIS-C. In fact, there is no case report of MIS-C associated with COVID-19 vaccination; COVID-19 vaccination even protects against rather than induces MIS-C [102]. These immunological features of MIS-C were studied in the early waves of COVID-19 pandemic with the Wuhan strain and the Alpha variant. The effects of the Delta and Omicron variants on innate and adaptive immune responses in patients with COVID-19, MIS-C, and MIS-A remains to be determined. The complex associations suggest the development of MIS-C and MIS-A might require not only single gene mutation, but also combining two or more variants of immune genes.

## 6. Immunopreventive Tactics of Severe Non-Pulmonary COVID-19 Diseases

A non-pulmonary COVID-19 is usually induced by SARS-CoV-2 infection, followed by viral spreading while innate immunity cannot limit the virus replication. Hosts with certain genetic variants might cause malignant fever and mitochondrial (Mito.) dysfunction associated with hyperinflammation resulting in ANE in deep brain regions (Figure 3, left panel). The viral spreading can be interrupted by adaptive immunity such as T cell immunity resulting in clearance of viral spreading for recovery. Hosts with altered immune reactions could cause cytokine storm or T cell over-activation such as production of IFNγ and hemophagocytosis under impairment of T regulatory cells (Treg), resulting in MIS-C or MIS-A (Figure 3, right panel). Based on the altered infection immunity specified, we could raise six immunological tactics to prevent the severe non-pulmonary diseases as follows:

### 6.1. Virus Neutralization by Active and Passive Immunization

Given the safety consideration, vaccine availability for children, particularly for infants, is authorized late, so the herd immunity in adults is important for protecting infants and children from infections. Similarly, passive immunization with specific monoclonal antibodies is also authorized late for children and infants. Although the SARS-CoV-2 has evolved into immune escape of vaccines and monoclonal antibodies, the current vaccines remain effective on reducing morbidity and mortality. In the COVID-19 pandemic, family contacts are more important than school contacts for the social distancing in family is much shorter than those in schools, especially while a school policy restricts affected children from school. The herd immunity in the household (family and daycare centers) would be important for providing protection of children from COVID-19. In addition, since application of monoclonal antibodies and anti-virus drugs are limited for small children, it is also important to prepare a convalescent plasma with high neutralizing antibody titers directed against homologous virus variant for preventing infants and children with immunodeficiency or immunocompromised situation from severe COVID-19.

For newborns and premature babies, the prevention of COVID-19 should be started from maternal vaccination. Pregnant mothers should have a full course of vaccination with the boosting dose in the third trimester, which will ensure the effective neutralizing antibodies for newborns and infants via transplacental transportation of IgG. Thus, next generation of COVID-19 vaccines should be made available for pregnant women as fast as possible.

### 6.2. Anti-Virus Replication by Small Molecules

The SARS-CoV-2 Omicron variants have evolved to immune escape of immunization and high transmission rate through upper but not lower respiratory tract infection [103]. Antiviral drugs have been shown to block viral replication and to reduce viral load, resulting in decrease of morbidity and mortality. Currently, three anti-virus drugs including Paxlovid (nirmatrelvir and ritonavir), Lagevrio (molnupiravir), and Veklury (remdesivir) maintain actively against different SARS-CoV-2 variants including Omicron BA.2 and BA.5 [104]. Remdesivir is the one with emergency use authorization to children under age 12, based on a bridge clinical trial with 53 children between ages one month and twelve [105]. For children with ANE, the immunomodulation and supplementation of mitochondrial cocktails within 24 h are shown to rescue certain portion of children with better outcomes [84]. There is no report regarding whether remdesivir could rescue children with ANE.

### 6.3. Early Reduction of Viral Load by Monoclonal Antibodies or Trispecific DARPin

In addition to antiviral drugs, there are also other ways to reduce viral load and viral spreading. Most of the monoclonal antibodies used to reduce viral load in SARS-CoV-2 Alpha or Delta variants are no longer used for Omicron variants [103,106]. Currently, Bebtelovimab is the only monoclonal antibody authorized to treat Omicron patients with immunocompromised conditions in non-hospitalized patients [107]. It is not guaranteed that novel Omicron subvariants wouldn’t develop immune escape of the Bebtelovimab and beyond. Another way to reduce viral load is to use Ensovibep, which is a trispecific DARPin designed to bind serum albumin and three domains of SARS-CoV-2 spike antigen [108]. Ensovibep appeared to lower the risk of hospitalization, emergency room visits, or death from COVID-19 by 78% [109]. Hopefully, this novel design available for blocking viral entry without immune escape, reducing viral load, and decreasing hospitalization in adults and children.

### 6.4. Targeting Altered Immune Reaction and Cytokine Storm

In early waves of the COVID-19 pandemic, adult respiratory distress syndrome (ARDS) due to cytokine storm was the main cause of morbidity and mortality. Tocilizumab (Actemra), an anti-IL-6 receptor antibody, has been shown to reduce mortality in hospitalized patients in the first 2 days of ICU admission [110]. The FDA has also approved Olumiant (baricitinib), a Janus kinase inhibitor (JAKi), for treating severe COVID-19 patients with cytokine storm [111]. Tocilizumab can be used in children under age 12, but not yet Olumiant. Another disease modifying agent for cytokine storm of COVID-19 is Sabizabulin, which is a microtubule inhibition agent useful for suppressing leukocyte activation and reveals a good clinical efficacy for severe COVID-19 [112].

### 6.5. Immunoregulation of Inflammatory Non-Pulmonary COVID-19 

Since inflammatory non-pulmonary COVID-19 diseases are quite different from those in pulmonary COVID-19, different immunopathogenesis of COVID-19 diseases require different immunomodulations. For those with ANE, early pulse of methylprednisolone and hypothermia therapy should be instituted within 24 h [83,85]. Moreover, patients with ANE related to fever-sensitive or mitochondrial dysfunction should be administered with mitochondrial cocktails including biotin, thiamine, L-carnitine, and CoQ10 [55,84,85,86]. For those with post-infectious autoimmunity in children such as MIS-C, IVIG and pulse methylprednisolone are the mainstream of medications. In addition, the cell therapy with mesenchymal stem cells (MSCs), which can enhance regulatory functions of Treg cells, has been shown promising on rescue of COVID-19 patients with ARDS [113,114,115]. Whether the MSCs therapy could rescue patients with ANE deserves further studies.

### 6.6. Fever Control and Preservation of Mitochondrial Functions

One possible mechanism for infection associated ANE is depletion of energy resulting from fever induced mitochondrial dysfunction with oxidative stress and hyperinflammation [56,94,95,96]. Since infants and young children with respiratory infections tend to have fever, small children carrying fever-sensitive or metabolism-defective genes such as CPT2 and SLC19A3 could compromise mitochondrial functions of neuron cells on transportation of thiamine or carnitine, resulting in hyperinflammation due to oxidative stress and deprivation of energy production. Prevention of febrile infection associated ANE would require early detection of fever-sensitive genetic variants for avoiding infection, and early administration of nutrient supplements of mitochondrial metabolism [84,85,86].

## 7. Conclusions

An emerging infection usually breaks out in a population who have naïve immunity to the pathogen. In the early pandemic, the quarantine regulations including isolation, face masks, social distancing, and lockdown are important to block the pandemic until herd immunity is built up. Ideally, prophylactic vaccines are the best strategy to block pandemic. It is fast enough that humankind has DNA, RNA, and protein vaccines of COVID-19 made useful for preventing people from hospitalization and mortality. Unfortunately, ongoing SARS-CoV-2 variants reveal immune escape from the vaccines made in original strain and subvariants. Several waves of outbreaks have occurred in different countries depending on dynamic interactions among viral variants, quarantine isolation, lockdown, active immunization, passive immunization, and early anti-virus medications. The greater the later virus variants the higher the transmission rate (reproduction number) is. The Omicron BA.5 variant after the Alpha, Beta, Gamma, and Delta had the highest transmission rate and immune escape of vaccines. The late emergency use authorization of vaccines for children has resulted in low levels of neutralizing antibody titers, higher viral load, and more non-pulmonary diseases. This situation would make patients with immunodeficiency just as bad as hyperinflammation leading to severe non-pulmonary inflammatory diseases such as ANE, MIS-C, and MIS-A (Table 1; Figure 3). Fortunately, children receiving mRNA vaccines appeared to reduce occurrence of MIS-C. Whether the mRNA vaccines protect children from ANE remains to be determined. Moreover, adolescents and adults could have ANE or MIS-A, which could cause a higher mortality, requiring early recognition and early treatment to reduce the mortality. The combined treatment of IVIG and methylprednisolone with and without cardiovascular supports could effectively rescue the patients with MIS-C or MIS-A; however, it remains unclear what regimen is suitable to rescue patients with ANE because of its elusive immunopathogenesis of hyperinflammatory mechanisms. The prevention of severe ANE might not be rescued by early use of anti-virus agents, but rather early use of mitochondrial cocktails and immunomodulators. 

## Figures and Tables

**Figure 1 ijms-23-14157-f001:**
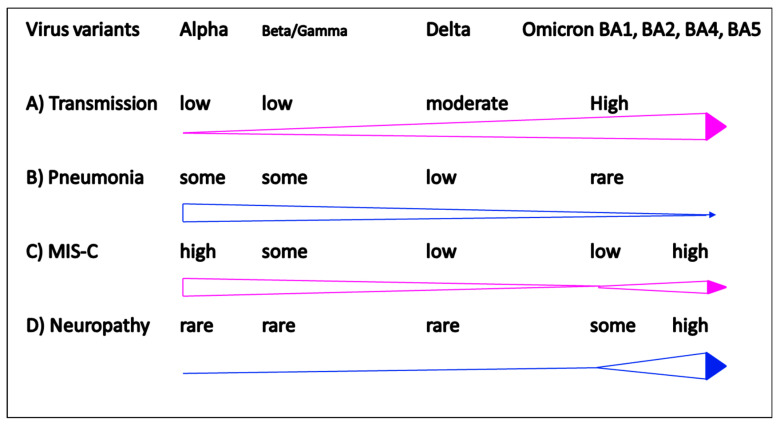
Clinical features in different pandemic waves of COVID-19. (**A**) Increase of transmission rate from Alpha, Beta, Delta to Omicron; (**B**) decrease of low respiratory tract pneumonia from Alpha, Beta, Delta to Omicron; (**C**) biphasic increase of MIS-C; (**D**) Increase of pediatric encephalopathies in Omicron pandemic.

**Figure 2 ijms-23-14157-f002:**
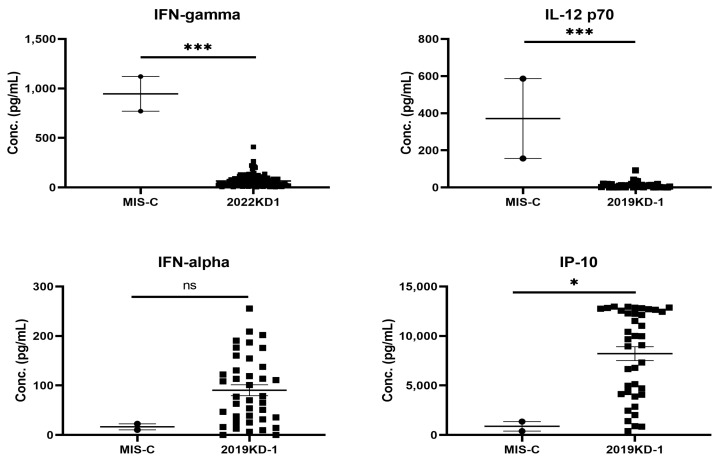
The skewed profiles of Th1 cytokines between patients with MIS-C and KD, showing significantly higher IL-12 and IFNγ levels in MIS-C (*n* = 2) than KD (*n* = 80) children (*** *p* < 0.001, Mann Whitney U test), but proinflammatory mediators, IP-10 and IFNα levels, were significantly lower in MIS-C than KD children (* *p* < 0.05, Mann–Whitney U test).

**Figure 3 ijms-23-14157-f003:**
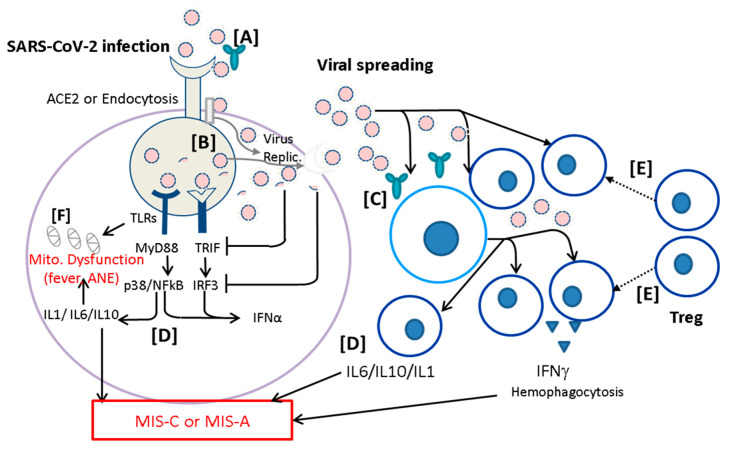
Prevention of severe non-pulmonary COVID-19 diseases. A COVID-19 disease is usually induced by SARS-CoV-2 infection, followed by viral spreading while innate immunity cannot limit the virus replication (Virus Replic.). Patients with certain genetic variants might cause malignant fever and mitochondrial (Mito.) dysfunction, associated with hyperinflammation resulting in acute necrotizing encephalopathy (ANE) in deep brain regions. The viral spreading can be interrupted by adaptive immunity such as T cell immunity resulting in recovery. Altered immune reactions could lead to (→) cytokine storm or T cell overactivation such as production of IFNγ and hemophagocytosis under impairment of T regulatory cells (Treg), resulting in MIS-C or MIS-A. Based on the altered infection immunity proposed here, we could prevent the severe non-pulmonary diseases as follows: [A] virus neutralization by active and passive immunization for limiting viral transmission by effective neutralizing antibodies; [B] anti-virus replication (Virus Replic.) by small molecules, such as paxlovid or remdesivir; [C] early reduction of viral load by monoclonal antibodies or trispecific DARPin; [D] targeting altered immune reaction and cytokine storm by inhibition of leukocyte activation; [E] immunoregulation (⸽) of altered inflammation by enhancement of regulatory T cells (Treg) through administration of IVIG, methylprednisolone, or mesenchymal stem cells; and [F] avoiding mitochondrial (Mito.) dysfunction by controlling fever and early supplying mitochondrial cocktails including biotin, thiamine, L-carnitine, and CoQ10.

**Table 1 ijms-23-14157-t001:** Immunopathogenesis of non-pulmonary COVID-19 diseases.

Disease	Aseptic Meningoencephalitis	Acute Necrotizing Encephalitis	Multisystem Inflammatory Syndrome in Children
Immature immunity	<5 Y, low antibody and high viral load	0.5~8.5 Y, hyperinflammation	2~18 (8.5) Y, post-infectious autoimmune
Infection-associated hyperinflammation	Viral invasion or infection-induced interruption of BBB	Fever or virus sensor defect linked to altered metabolism and immunity	Autoimmune vasculitis and thrombosis mediated by Th1 cytokine storm
Race Genetics	All races Developmental delay of immunity	Asians *RANBP2*, *CPT2*, *SLC19A3*, *SCN1A*	Hispanics, Asians Immune/phagocytosis genes, and TRBV11-2
Immunotherapies	Active immunization Passive immunization	Early immunomodulators Mitochondrial cocktails	Early immunomodulators Anti-thrombosis Rx.

Abbreviations: Y, year of age; BBB, blood–brain barrier; *RANBP2*, RAN binding protein 2; *CPT2*, carnitine palmitoyltransferase II; *SLC19A3*, solute carrier family 19 member 3; *SCN1A*, sodium voltage-gated channel alpha subunit 1; TRBV11-2, T cell receptor beta variable 11-2; Rx, treatment.

## Data Availability

The data included in the table and figures were summarized from previous publications of the literature on non-pulmonary COVID-19 diseases.

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
