# Peer review of "Pathogenesis and Preventive Tactics of Immune-Mediated Non-Pulmonary COVID-19 in Children and Beyond"

_ijms, 2022, doi:10.3390/ijms232214157_

Round 1
Reviewer 1 Report
Chi et al in the article titled `Pathogenesis and preventive tactics of immune -mediated non-pulmonary COVID-19` describe the after infection complications and also lists possible immunopreventive tactics for severe non-pulmonary COVID-19 diseases.
The article is well structured and written and the table and figures are of a good quality.
The only minor suggestion would be not to have so many repetitive sentences because it distracts a bit from the topic, but maybe that is just a writing style.
Also, the article should maybe have a small subtitle to stress the importance of the topic regarding infants and small children, since the major part of the article is dedicated to this age and partially to adults, especially in the MIS-A description.
Author Response
Responses to Reviewer-1’s suggestions
1. The repetitive sentences among different sections have been made non-redundant and logic precision as marked in red ink.
2. The subtitle “in children and beyond” has been added in the title on page 1. To fit in the subtitle, we also made it catchier in the Abstract (lines 30-34), Table 1, Figure 1, and Conclusion (lines 499-501).
Thank you very much for your nice comments by which we have made the article look upon improved and suitable for consideration of publication.
Reviewer 2 Report
1, What’s the difference in immunopreventive tactics between non-pulmonary COVID-19 and pulmonary COVID-19 diseases?
2, Statements in the following pages may require to be marked with clear references.
Page 3 line 136-137.
Page 5 line 243-245.
Page 8 line 370-378.
Page 8 line 382-401.
3,Please describe the literature search strategy and the inclusion criteria.
4, How about the heterogeneity of the diagnostic standard of ME, ANE, and MIS-C, respectively, among the literature?
Author Response
Responses to Reviewer-2’s suggestions
- In the introduction (page 4, lines 71-75), we described immunopreventive tactics for early prevention and treatment of the potentially fatal non-pulmonary COVID-19 different from conventional pulmonary COVID-19, based the immunopathogenesis of non-pulmonary COVID-19 different from pulmonary COVID-19. We also summarized the pathogenesis of different non-pulmonary COVID-19 diseases into a structured table describing different immunopathogenesis, infection-associated hyperinflammation, race and genetics, and immunotherapies for non-pulmonary COVID-19 (Table 1). We finally drew a figure depicting the immunopreventive tactics for non-pulmonary COVID-19 by active immunization, passive immunization, anti-leukocyte activation, anti-cytokine storm and correction of metabolism, which are different from the conventional anti-virus medication in pulmonary COVID-19. This is described in the revised manuscript page 5, lines 112-120.
- Clear references for the statements derived from literature have been precisely cited after each statement.
The sentences at Page 3 lines 136-137 have been corrected in the revised article (page 3, lines 67-71).
The sentences at Page 5 line 243-245 have been corrected in the revised article (page 7, lines 164-167).
The sentences at Page 8 line 370-378 have been corrected in the revised article (page 9, lines 228-230).
The sentences at Page 8 line 382-401 have been corrected in the revised article (page 9, lines 230-234).
- Methodologies to do structured literature search for non-pulmonary COVID-19 have been added to the 2nd section of the review articles (page 4, lines 77-120).
- The heterogeneity of the diagnostic criteria of ME, ANE and MIS-C have been defined in the Method section (page 5, lines 95-99).
Thank you very much for your nice comments by which we have made the article greatly improved.
Reviewer 3 Report
In this manuscript titled “Immunopathogenesis and preventive tactics of inflammatory non-pulmonary COVID-19”, the authors have summarized the evolution of COVID-19 from mild diseases in early pandemic waves to a higher severity and fatality of hyperinflammatory non-pulmonary diseases in the later waves of the pandemic. The manuscript is not well-structured. The manuscript can be considered after making the following amendments:
· The abstract section should be refined to make it short and catchier.
· The introduction section is missing.
· Check and revise the serial numbers for the heading and subheading to make it inconvenient for the reader to understand.
· Methodology section is missing.
· There are a few typos and unclear sentences which should be rectified.
· Rewrite the sentence from lines 64-67 for more clarity “More importantly, adolescents and adults are not spared of fatal encephalopathies such as acute necrotizing encephalopathies (ANE) [18-20], and multisystem inflammatory syndrome in adults (MIS-A), which has even caused a higher mortality than that in MIS-C [21-23].”
· Rewrite the statement from lines 358-361 “It is, however, unlikely reasonable to recognize the S epitope associated autoreactive clone(s) involved in the MIS-C because if the epitope in S antigen could induce autoreactive T cell expansion, the COVID- 19 vaccines made in full length of Spike antigen could also induce the MIS-C.” It’s hard to read.
·
Author Response
Responses to Reviewer-3’s comments
- The abstract section has been made it more concise (from 196 words to 162 words), and made it catchier by integrating “These life-threatening non-pulmonary COVID-19 such as acute necrotizing encephalopathies (ANE) or multisystem inflammatory syndrome in children (MIS-C) are more prevalent in children. However, the mortality of multisystem inflammatory syndrome in adults (MIS-A) is much higher than that of MIS-C although the incidence of MIS-A is lower.” into the abstract.
- The introduction section has been re-organized in page 3, lines 43-75.
- The heading and subheading in the text have been made clear by beginning at the heading with boldface, followed by Italic subheading and subsequently Arabic numerals for different items.
- A Methodology section has been added in page 4, lines 77-120.
- Those unclear sentences have been corrected or rectified. 5.1 The sentence from lines 64-67 has been rectified in page 3, lines 67-71 5.2 The sentence from lines 358-361 has been rectified in page 15, lines 378-383. 5.3 Other sentences corrected are marked in red ink.
Thank you very much for your nice comments by which we have made the article look upon favorable for consideration of publication.
Round 2
Reviewer 3 Report
I think the authors have adequately addressed the comments in the revised version of the manuscript. Therefore, I have no further comments.